# Birch Sap Preserves Memory Function in Rats by Enhancing Cerebral Blood Flow and Modulating the Presynaptic Glutamatergic System in the Hippocampus

**DOI:** 10.3390/ijms26115009

**Published:** 2025-05-22

**Authors:** Chien-Fen Huang, Tzu-Kang Lin, Chia-Chuan Chang, Ming-Yi Lee, Ching-Yi Lu, Chi-Feng Hung, Su-Jane Wang

**Affiliations:** 1Ph.D. Program in Pharmaceutical Biotechnology, Collage of Medicine, Fu Jen Catholic University, New Taipei 24205, Taiwan; fanny@dradvice.com; 2School of Medicine, Fu Jen Catholic University, New Taipei 24205, Taiwan; tklin@gmail.com; 3Department of Neurosurgery, Fu Jen Catholic University Hospital, Fu Jen Catholic University, New Taipei 24205, Taiwan; 4School of Pharmacy, National Taiwan University, Taipei 106319, Taiwan; chiachang@ntu.edu.tw (C.-C.C.); dl0423202@ntu.edu.tw (C.-Y.L.); 5Department of Medical Research, Far-Eastern Memorial Hospital, New Taipei 22060, Taiwan; mingyi.lee@gmail.com; 6School of Pharmacy, Kaohsiung Medical University, Kaohsiung 80708, Taiwan; 7Research Center for Chinese Herbal Medicine, College of Human Ecology, Chang Gung University of Science and Technology, Taoyuan 33303, Taiwan

**Keywords:** birch sap, memory function, glutamate release, synaptic proteins, synaptosome, hippocampus

## Abstract

As the average age of the population increases, memory impairment has become an increasingly prevalent issue. This study investigates the effects of 14 days of oral birch sap administration on memory functions in healthy rats using the Morris water maze (MWM) test and explores the underlying mechanisms. A compositional analysis revealed that birch soap is rich in polysaccharides, specifically a low-molecular weight polysaccharide (MW 1.29 kDa), and exhibits no hepatotoxicity or renal toxicity at the tested dose. The results from the MWM test demonstrated that the time and distance required to reach the platform were significantly shorter in the birch sap-treated group compared to the control group, suggesting that birch sap supports memory preservation. Moreover, rats treated with birch sap showed improved cerebral blood flow compared to the control rats. Additionally, in hippocampal nerve terminals (synaptosomes), rats treated with birch sap exhibited a significant increase in evoked glutamate release, as well as elevated levels of presynaptic proteins, including vesicular glutamate transporter 1 (VGluT1), synaptophysin, synaptobrevin, synaptotagmin, syntaxin, synapsin I, and the 25 kDa synaptosome-associated protein (SNAP-25). Transmission electron microscopy also revealed a notable increase in the number of synaptic vesicles in hippocampal synaptosomes of the birch-sap-treated rats. These findings suggest that birch sap enhances hippocampal presynaptic glutamatergic functions and cerebral blood flow, contributing to its memory-preserving effects in rats.

## 1. Introduction

As the global population ages, the prevalence of neurodegenerative disorders, such as Parkinson’s disease and Alzheimer’s disease (AD), is rising, with memory loss being a prominent symptom of these conditions [1,2]. Currently, drug treatments for memory impairment are often limited in effectiveness and accompanied by notable side effects [3]. As a result, natural products, such as medicinal plant extracts, have gained significant attention for their potential to support memory functions, due to their diverse active compounds and relatively mild side effects [4,5,6,7].

Birch sap is a clear or pale yellow liquid extracted from the bark of birch trees in the Betula genus of the Betulaceae family. Rich in essential nutrients, including sugars, amino acids, vitamins, and minerals, birch sap is commonly used in the food and beverage industry [8,9]. Birch sap has been traditionally used to treat various conditions, including hepatitis, skin rashes, intestinal worms, and scurvy [10]. However, no studies have yet explored its effects on the central nervous system (CNS), particularly on the glutamate system. Glutamate, the primary excitatory neurotransmitter in the mammalian CNS, plays a crucial role in memory maintenance [11]. Numerous studies have shown that reductions in brain glutamate levels impair memory and learning [12,13,14,15]. Therefore, strategies aimed at enhancing glutamatergic neurotransmission, such as increasing synaptic glutamate release, may help prevent memory decline [16,17].

In this study, we (i) evaluated the effects of two weeks of oral birch sap administration on memory in healthy rats using the Morris water maze (MWM) test, (ii) investigated glutamatergic functions, including synaptic glutamate release and the levels of synaptic proteins, in synaptosomes isolated from the hippocampus—a brain region critical for memory functions [17,18,19], and (iii) assessed the impact of birch sap on rat body weights, as well as liver and kidney morphology. The results of this study may contribute to exploring the potential application of birch sap as a food ingredient or supplement to help prevent memory impairment.

## 2. Results

### 2.1. Characterization of Oligosaccharides from Birch Sap

After freeze-drying 50 mL of birch sap, a total of 23.5 g of solid polysaccharides was successfully obtained. The analysis of the monosaccharide composition, as shown in Figure 1A and detailed in Table 1, indicates that birch sap is predominantly composed of fucose, glucose, and fructose. The relative proportions of these monosaccharides were determined to be approximately fucose/glucose/fructose = 3/5/8. This composition suggests that fructose is the most abundant monosaccharide in birch sap, followed by glucose, with fucose being present in comparatively lower quantities. In addition to the monosaccharide composition, the molecular weight analysis (shown in Figure 1B) revealed that the molecular weight of birch sap is 1.29 kDa. The 1H NMR spectrum of birch sap displayed two anomeric protons at δH 5.13 and 4.45. The signals at δH 5.13 and δH 4.45 appeared as doublets in a ratio of 100:124, and the two monosaccharide residues exhibited α-(^3^J_1,2_ = 3.4 Hz) and β-(^3^J_1,2_ = 7.7 Hz) anomeric configurations, respectively (Figure 1C).

### 2.2. Behavioral Evaluation of Spatial Memory in Rats

Figure 2 shows the study design. Birch sap was orally administered to rats at doses of 3, 6, and 10 mL/kg once daily for 14 days. The MWM test was used to evaluate the effect of birch sap on the rats’ spatial memory. As shown in Figure 3A,B, the escape latency (time to find the platform) and movement distance in the birch sap groups were significantly shorter compared to the control group (latency to target: F(3.29) = 16.6, *p* < 0.0001; distance to target: F(3.29) = 17.5, *p* < 0.0001), indicating that the rats in the birch-sap-treated groups had a stronger memory of the platform location, demonstrating enhanced spatial memory abilities. Since 10 mL/kg of birch sap was more effective than 3 mL/kg and 6 mL/kg in preserving memory in rats (*p* < 0.05), a dose of 10 mL/kg was selected for the subsequent assays.

### 2.3. Evaluation of Cerebral Blood Flow in Rats

Cerebral blood flow is strongly associated with memory functions, and a reduction in blood flow to the brain can lead to memory impairment [20,21]. Therefore, we assessed the effect of birch sap on cerebral blood flow in rats. As shown in Figure 4A,B, cerebral blood flow was significantly higher in the birch-sap-treated group compared to the control group (t(8) = 5.6, *p* < 0.0001). This result suggests that birch sap may enhance cerebral blood flow, which is linked to the observed memory preservation.

### 2.4. Assessment of Glutamate Release in Hippocampal Synaptosomes

Since the glutamatergic system plays a crucial role in memory functions [11], we studied the synaptic glutamate release in hippocampal synaptosomes from control and birch-sap-treated rats. As shown in Figure 5A,B, no significant differences were observed in basal glutamate release from hippocampal synaptosomes between the control and birch-sap-treated groups (*p* > 0.05). However, the birch-sap-treated group exhibited a significant increase in 4-aminopyridine (4-AP)-evoked glutamate release from hippocampal synaptosomes compared to the control group (t(8) = 12.9, *p* < 0.0001). This suggests that enhanced presynaptic glutamate release in the hippocampus may contribute to the memory-preserving effects of birch sap observed in this study.

### 2.5. Assessment of Synaptic Protein Expression in Hippocampal Synaptosomes

To elucidate the potential molecular mechanisms by which birch sap enhances presynaptic glutamatergic functions, we assessed the levels of presynaptic proteins in hippocampal synaptosomes, including VGlut1 (a marker of glutamatergic terminal), synaptophysin, synaptotagmin, synaptobrevin, syntaxin, SNAP-25, synapsin-1, and p-synapsin I. As shown in Figure 6A–H, representative Western blot bands and statistical analysis revealed that the protein expression levels of VGlut1, synaptophysin, synaptotagmin, synaptobrevin, syntaxin, SNAP-25, synapsin I, and p-synapsin I were significantly higher in the hippocampal synaptosomes of the birch-sap-treated group compared to the control group (VGlut1, t(8) = 13.7, *p* < 0.0001; synaptophysin, t(8) = 10.3, *p* < 0.0001; synaptotagmin, t(8) = 16.7, *p* < 0.0001; synaptobrevin, t(8) = 16.7, *p* < 0.0001; syntaxin, t(8) = 19.3, *p* < 0.0001; SNAP-25, t(8) = 15.6, *p* < 0.0001; synapsin I, t(8) = 14.3, *p* < 0.0001; *p*-synapsin I, t(8) = 27.6, *p* < 0.0001).

### 2.6. Assessment of Synaptic Vesicle Numbers in Hippocampal Synaptosomes

In Figure 7A, the ultrastructure of hippocampal synaptosomes and the number of synaptic vesicles were observed using transmission electron microscopy (TEM). Consistent with previous studies [22,23], synaptosomes appeared as round, membrane-bound structures containing mitochondria and an abundance of synaptic vesicles. Figure 7B shows significant differences in the number of synaptic vesicles between the control group and the birch-sap-treated group (t(6) = 4.7, *p* < 0.001).

### 2.7. Evaluation of Birch Sap Effect on Body Weight, Liver, and Kidney Morphology in Rats

Birch sap at a dose of 10 mL/kg was orally administered to rats once daily for 14 days. Throughout the experimental period, no animal deaths occurred in any group. Normal body weight gain was observed, with no significant differences between the groups (t(26) = 0.15, *p* = 0.88; Figure 8A). Additionally, hematoxylin-eosin (H&E) staining and the pathologist’s report indicated that liver and kidney tissues were entirely normal in both the control group and the birch-sap-treated group. No signs of necrosis, degeneration, or inflammation were observed in either group, suggesting that birch sap does not induce any noticeable toxic effects on the liver and kidney in rats (Figure 8B).

## 3. Discussion

Developing memory-enhancing drugs is a global challenge, and the search for effective treatments continues. This research explored the promising potential of birch sap in memory preservation, given its various health benefits [10]. In this study, we evaluated the effect of birch sap on memory functions in rats using the MWM test and investigated its possible underlying mechanisms, particularly in the glutamatergic system, which is closely associated with the memory maintenance [11]. This investigation could provide a theoretical basis for the use of birch sap as a therapeutic agent and in functional foods in slowing memory decline.

Birch sap is water-soluble, and our HPLC quantification of birch sap identified polysaccharides as a major compound, consistent with previous studies [10,24]. Therefore, polysaccharides could serve as a promising quality control marker for birch sap raw material powder, helping to address challenges in standardizing birch sap for research and medicinal applications. In addition, in the present experiment, birch sap was orally administered to rats at a dose of 10 mL/kg once daily for 14 days. During the experiment, no significant differences in body weight were observed between the control and birch sap treatment groups. Furthermore, our results indicated that birch sap administration did not cause significant changes in the morphology of the liver or kidneys in rats. These findings support the safety of birch sap for daily use.

In the present study, we assessed the effect of birch sap on spatial memory using the MWM test, the most widely accepted model for evaluating spatial learning and memory in rodents. In the MWM test, a lower escape latency score indicates enhanced spatial learning and memory [25]. We found that the oral administration of birch sap for 14 days significantly decreased escape latencies compared to the control group. The shorter escape latencies suggest that birch sap may enhance memory retention in healthy rats at this dose. The observed preservation of memory function mediated by birch sap may be attributed to its polysaccharide content, as characterized by the monosaccharide composition (Figure 1A and Table 1), molecular weight (Figure 1B), and ^1^H NMR spectroscopic analysis (Figure 1C). Plant-derived polysaccharides have been shown to improve memory functions [26]. For example, an active polysaccharide from *Nicotiana* leaves mitigates memory deficits in mice by regulating a gut–brain pathway involving the microbiota, IL-6, and H3K27me3 [27]. Additionally, polysaccharides from buckwheat and *Pseudostellaria heterophylla* retore memory and learning ability through anti-inflammatory effects and modulation of the gut microbiota in aluminum chloride (AlCl_3_)-treated rats and 5 × FAD model mice, which are commonly used AD models [28,29]. Additionally, memory impairment is frequently linked to reduced cerebral blood flow [20,21]. In this study, a significant increase in cerebral blood flow was observed in the birch sap-treated rats, suggesting a potential causal relationship between enhanced cerebral blood flow and the memory-preserving effects of birch sap. However, how birch sap modulates cerebral blood flow was not elucidated in this study and should be investigated further.

Considering the observed memory preservation, the current study focused on changes in glutamate, a major neurotransmitter involved in memory formation and storage [11,30]. Studies have shown that reduced glutamate levels in brain regions, particularly in the hippocampus, are closely linked to memory impairment [17,30,31]. In the present study, we found that 4-AP-evoked glutamate release in the hippocampal nerve terminals (synaptosomes) of the birch-sap-treated group was higher than in the control group. Furthermore, a significant increase in the expression levels of VGluT1 was observed in the hippocampal synaptosomes of the birch-sap-treated rats. VGluT1 is located on synaptic vesicles and mediates the uptake of glutamate into these vesicles within adult hippocampal neurons [32]. Its upregulation leads to an increase in the number of vesicles per terminal, higher glutamate release probability, enhanced excitatory postsynaptic currents, increased long-term potentiation (LTP), and ultimately, memory enhancement [12,33]. In addition to VGlut1, the expression of glutamate-release-related synaptic proteins, including SNAP-25, synaptobrevin, p-synapsin I, and syntaxin, was significantly increased in in the hippocampal synaptosomes of birch-sap-treated rats. Notably, the increased phosphorylation of synapsin I promotes synaptic vesicle trafficking and enhances the size of the releasable vesicle pools [34,35]. Consistent with this, TEM results also revealed a significant increase in the number of synaptic vesicles in the hippocampal synaptosomes of these rats. These findings suggest that birch sap enhances evoked glutamate release by increasing synaptic proteins and vesicles in the hippocampal synaptosome, which may contribute the observed memory retention in rats during the MWM tasks. This speculation is supported by evidence showing that high levels of presynaptic proteins, including synaptophysin, SNAP-25, syntaxin, and synaptobrevin, are associated with better cognitive performance and a lower risk of dementia in older adults [36,37,38,39,40]. Overall, the presynaptic glutamatergic changes represent a significant molecular factor contributing to the observed preservation of memory functions in rats treated with birch sap. To further support this hypothesis, future studies should investigate how birch sap increases the levels of synaptic proteins and vesicles in the hippocampal nerve terminals, thereby enhancing glutamate release and preserving memory functions.

The present study has several limitations. Firstly, only male rats were used in this study, and female rats were not assessed. Secondly, this study alone could not assess the effects on other neurotransmitter systems related to memory functions, such as the serotonergic, dopaminergic, and cholinergic systems [41,42,43]. Future studies in this area could help strengthen findings regarding the efficacy of birch sap in preserving memory functions. Thirdly, in the present study, birch sap was administered for a limited duration of two weeks. However, the long-term effects of birch sap ingestion remain unclear. Future studies involving prolonged administration are warranted to evaluate potential behavioral or neurochemical alterations resulting from chronic exposure. Finally, other models of memory impairment, such as AD or aging animal models, should be evaluated to enhance the clinical relevance of birch sap’s effects.

## 4. Materials and Methods

### 4.1. Materials

Birch sap was sourced from Chihiro Kawase, Forest Works (Hokkaido, Japan). 4-AP was purchased from Tocris (Bristol, UK). Anti-synaptophysin (#36406S), anti-synapsin I (#5297S), and anti-β-actin (#3700S) were purchased from Cell Signaling (Danvers, MA, USA). Anti-synaptotagmin (#ab13259), anti-synaptobrevin (#ab18013), anti-syntaxin (#ab188583), and anti-synaptosomal-associated protein 25 kDa (SNAP 25, #ab41455) were purchased from Abcam (Cambridge, UK). Anti-phospho-synapsin I (Ser 603) (#GTX82589) and anti-horseradish peroxidase-conjugated secondary antibodies (#GTX213110-01, #GTX213111-01) were purchased from Gentex (Zeeland, MI, USA). Nicotinamide adenine dinucleotide phosphate (NADP^+^), glutamate dehydrogenase (GDH), and other chemical reagents were purchased from Sigma-Aldrich (St. Louis, MO, USA).

### 4.2. Analysis of Birch Sap Polysaccharides

#### 4.2.1. The Freeze-Dried Procedure for the Liquid Polysaccharide

The 50 mL of liquid polysaccharide was placed at −80 °C overnight. Once the liquid polysaccharide had frozen into ice, the ice was freeze-dried to allow the water to sublimate, resulting in the formation of a solid polysaccharide.

#### 4.2.2. Molecular Weight Analysis

Polysaccharides were prepared at a concentration of 1 mg/mL in distilled water and analyzed using two high-performance size-exclusion chromatography (HPSEC) columns: G4000PWXL (7.8 × 300 mm) and G3000PWXL (7.8 × 300 mm). The analysis was conducted with a Viscotek model TDA-3-1 (Viscotek, Houston, TX, USA) relative viscometer, maintaining a flow rate of 0.5 mL/min, with deionized water as the eluent. A calibration curve was established using the Sodex P-82 series from Showa Denko America (New York, NY, USA) with different Pullulan standards, raffinose, and glucose.

#### 4.2.3. Monosaccharide Analysis

The monosaccharide composition of the polysaccharides was determined using high-performance anion-exchange chromatography (HPAEC) equipped with a pulsed amperometric detector (PAD) featuring a gold working electrode and a Carbopac PA-10 anion-exchange column (4.6 × 250 mm) from Dionex BioLC (Sunnyvale, CA, USA). For the analysis, 1 mg of each sample was hydrolyzed with 1.95 N trifluoroacetic acid (TFA) at 80 °C for 6 h. After hydrolysis, the residual TFA was removed under reduced pressure, and the resulting samples were dissolved in Milli-Q water. The aqueous solutions were filtered through a 0.22 μm filter before HPAEC analysis.

The mobile phase consisted of an isocratic solution of 18 mM NaOH, maintained at ambient temperature, with a flow rate set at 1.0 mL/min. The identification and quantification of monosaccharides were accomplished by comparing the analytical results with those of standard monosaccharides. A selection of nine monosaccharide standards—myo-inositol, sorbitol, fucose, galactosamine, glucosamine, galactose, glucose, mannose, and fructose—was obtained from Sigma-Aldrich Co. (St. Louis, MO, USA). Data collection and integration were performed using a PRIME DAK system from HPLC Technology Ltd. in the Macclesfield, UK.

### 4.3. Animals

Adult male Sprague–Dawley rats (*n* = 107; 7–8 weeks old; weight 160–200 g) were used in the experiments. Animals were housed in polypropylene cages, under standardized conditions of temperature (22 ± 2 °C) and humidity (70%), with a 12 h light/dark cycle and free access to water and food. This study was approved by the Animal Care Committee of Fu Jen Catholic University (approval number: A11319). All experimental procedures were conducted following the International Guidelines for Care and Use of Laboratory Animals.

### 4.4. Experimental Protocol

Rats were trained in the MWM test for 3 days. After training, they were randomly assigned to either the control group (receiving saline at 10 mL/kg/day via oral gavage for 14 days) and the birch sap treatment group (receiving birch sap at 3, 6, or 10 mL/kg/day via oral gavage for 14 days). Throughout the experimental period, all animals exhibited stable behavior and showed no overt signs of stress or agitation. The rats’ body weights were measured over 14 consecutive days (days 4 to 17). The MWA test was repeated on day 18, and the data were analyzed using the MWM video analysis system. On day 19, the rats were anesthetized with 3% isoflurane, and cerebral blood flow analysis was performed. Following the analysis, the rats were euthanized, and their brains were removed to collect the hippocampi for synaptosome preparation, which was used for glutamate release assays, Western blotting, and TEM. Additionally, the liver and kidneys were collected for H&E staining. The study design is summarized in Figure 2.

### 4.5. MWM Test

The MWM test was conducted to evaluate spatial learning and memory performance, as described in previous studies [22,44]. A circular pool with a diameter of 55 cm and a height of 25 cm was filled with opacified water to a depth of 20 cm, maintained at 25 ± 1 °C. The pool was divided into four quadrants, with the escape platform (10 cm in diameter) positioned at the center of one fixed quadrant for all trials. A video camera was placed above the tank to record the experiment. The MWM test included three days of training, followed by a test session on day 4. Training was conducted four times a day, and the escape latency time for each rat to go to the platform was measured within 120 s. Rats that reached the platform were allowed to remain there for 15 s. If the rats did not find the platform within the time limit, they were guided by the experimenter and allowed to remain on the platform for 30 s. The latency period for these rats was recorded within 120 s. The swimming path from the entry point to the hidden platform, escape latency, and movement distance within the coverage zone were recorded using a video-tracking system (Version 1.17, SINGA Technology Corporation, Taipei, Taiwan).

### 4.6. Cerebral Blood Flow Assessment

As previously described [45], cerebral blood flow in rats was assessed using a Laser Speckle Imaging System (RFLSI III, RWD, Shenzhen, China), which allows for the real-time monitoring of cerebral blood flow dynamics. Briefly, the rats were anesthetized with 3% isoflurane and secured in a stereotaxic frame. A midline incision was made on the skull to expose the calvaria, and a tampon was used to keep the area clean and dry during imaging. A laser was then directed at the skull, and high-resolution blood flow speckle images were captured using a CMSO camera (Brooklyn, NY, USA). Cerebral blood flow was continuously measured for 15 s with an observation height of 25 cm, a laser irradiation area of 1.7 cm, and an image matrix of 2064 × 1544. The resulting blood flow data were expressed in perfusion units.

### 4.7. H&E Staining

Liver or kidney tissues were fixed in 4% paraformaldehyde (PFA), then dehydrated through a graded series of alcohol solutions, and embedded in paraffin wax. A series of 5 μm paraffin sections was cut using a Leica rotation microtome and stained with H&E. Images of the stained sections were captured under a microscope at 400× magnification. Histological changes in the liver and kidney tissues were evaluated based on the following criteria: cytoplasmic color fading, vacuolization, nuclear condensation, nuclear fragmentation, nuclear fading, and erythrocyte stasis [46].

### 4.8. Preparation of Synaptosomes

Hippocampal synaptosomes were prepared as previously described [47,48]. Briefly, the hippocampus was homogenized in an ice-cold medium containing 0.32 M sucrose (pH 7.4), followed by centrifugation at 3000× *g* for 10 min at 4 °C. The supernatant was collected and centrifuged at 14,500× *g* for 12 min at 4 °C. The resulting pellet was resuspended and layered onto a discontinuous Percoll gradient (3%, 10%, and 23%) and then centrifuged at 32,500× *g* for 7 min at 4 °C. Synaptosomes, which formed between 10 and 23% of the Percoll gradient layers, were collected, washed twice with a HEPES buffer medium (HBM), and centrifuged at 27,000× *g* for 10 min. The synaptosomal pellet was resuspended in HBM, and the final protein concentration was adjusted to 0.5 mg/mL The synaptosomes were maintained at 4 °C throughout the procedure and used for glutamate release assays, TEM, and Western blot analyses.

### 4.9. Glutamate Release Assay

As previously described [47,49], synaptosomal pellets (0.5 mg protein) were resuspended in HBM containing 16 μM bovine serum albumin and incubated in a stirred, thermostated cuvette at 37 °C in a Perkin-Elmer LS-55B spectrofluorimeter (Springfield, IL, USA). After 3 min, NADP^+^ (2 mM), GDH (50 U/mL), and CaCl_2_ (1 mM) were added. After 10 min of incubation, 4-AP (1 mM) was added to stimulate glutamate release. Glutamate release was monitored by measuring the increase in fluorescence (excitation at 340 nm, emission at 460 nm), which corresponds to NADPH production through the oxidative deamination of released glutamate by GDH. Data were collected at 2 s intervals. A standard of exogenous glutamate (5 nmol) was added at the end of each experiment, and the fluorescence response was used to calculate the amount of released glutamate. The results are expressed as nanomoles of glutamate per milligram of synaptosomal protein (nmol/mg). Values presented in the text and shown in bar graphs represent the cumulative glutamate release after 5 min of depolarization with 4-AP and are expressed as nmol/mg/5 min.

### 4.10. TEM

TEM was performed as previously described [22,50]. Rat hippocampal synaptosomes were immersed in an electron microscope fixative solution for 1 day. The samples were then postfixed in 1% osmium tetraoxide for 2 h, followed by dehydration through a graded ethanol series, soaking, and embedding in pure epoxy resin. The samples were sectioned into 70 nm-thick slices and stained with uranyl acetate and lead citrate. Finally, the sections were examined under a TEM (JEM-1400, JEOL, Tokyo, Japan).

### 4.11. Western Blotting Analysis

As previously described [50], hippocampal synaptosomes were homogenized, and protein concentrations were determined using Bradford’s method. Equal amounts of protein (20 μg) were separated via sodium dodecyl sulfate (SDS)-polyacrylamide gel electrophoresis and transferred to nitrocellulose membranes (Amersham Biosciences, Amersham, UK) through electroblotting. The membranes were blocked with 3% bovine serum albumin in Tris-buffered saline (TBS) containing 0.05% Tween-20 (TBST) for 1 h at room temperature and then incubated overnight at 4 °C with primary antibodies. The antibodies used were as follows: anti-VGluT1 (1:5000), anti-synaptophysin (1:20,000), anti-synaptotagmin (1:1000), anti-synaptobrevin (1:800), anti-syntaxin (1:10,000), anti-synapsin I (1:50,000), anti-SNAP 25 (1:50,000), anti-phospho-synapsin I site-4, 5 (1:2000), anti-phospho-synapsin I site-3 (1:2000), and anti-β-actin (1:1000). The membrane was washed three times with TBST for 10 min each, followed by incubation with a horseradish-peroxidase-conjugated secondary antibody (1:5000) at room temperature for 90 min. The membranes were then incubated with an enhanced chemifluorescent substrate (Amersham Biosciences) and analyzed using ImageJ software (version 1.53k, Synoptics, Cambridge, UK).

### 4.12. Statistical Analysis

All statistical analyses were performed using GraphPad Prism software (version 7.0, Boston, MA, USA). Data normality was assessed with the Shapiro–Wilk test. Comparisons between two groups were analyzed using a two-tailed Student’s *t*-test. For comparisons among multiple groups, one-way analysis of variance (ANOVA) was used, followed by post hoc Tukey’s tests. Results are expressed as the mean ± standard error of the mean (SEM). A *p*-value less than 0.05 was considered statistically significant.

## 5. Conclusions

This is the first study to investigate the potential of birch sap as a treatment for memory preservation. Our findings are the first to show that birch sap can regulate cerebral blood flow and enhance hippocampal presynaptic glutamate functions, which may underlie its memory-preserving effects in rats (Figure 9). While these results suggest that birch sap could be used as a food ingredient or supplement to prevent memory loss, further clinical research is necessary to confirm its efficacy and safety in humans and to fully elucidate its mechanisms of memory preservation.

## Figures and Tables

**Figure 1 ijms-26-05009-f001:**
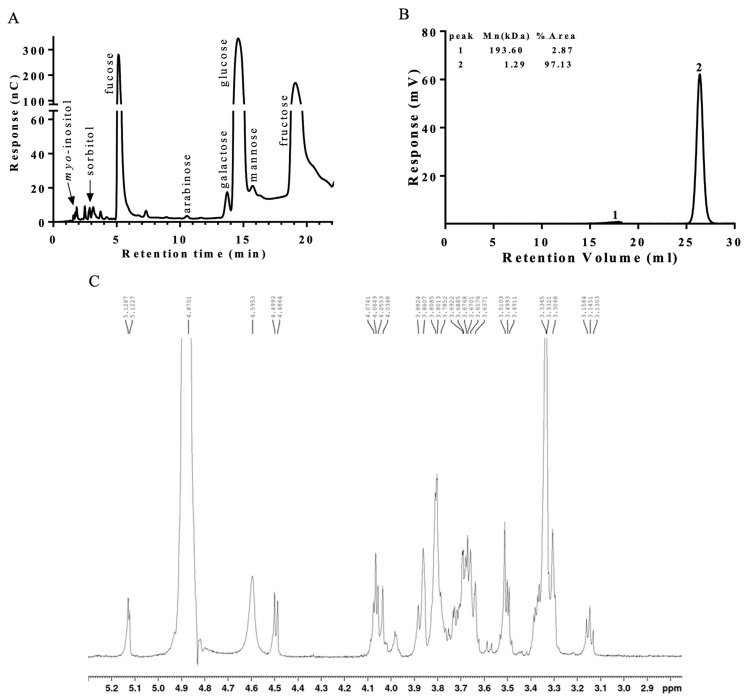
Chemical characterization of birch sap. (**A**) HPAEC-PAD chromatogram of birch sap. (**B**) Molecular weight analysis of birch sap. (**C**) ^1^H NMR spectrum of birch sap. (600 MHz, D_2_O).

**Figure 2 ijms-26-05009-f002:**
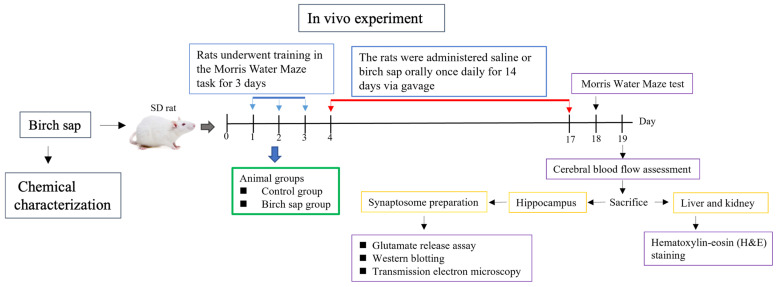
Schematic representation of the experimental design.

**Figure 3 ijms-26-05009-f003:**
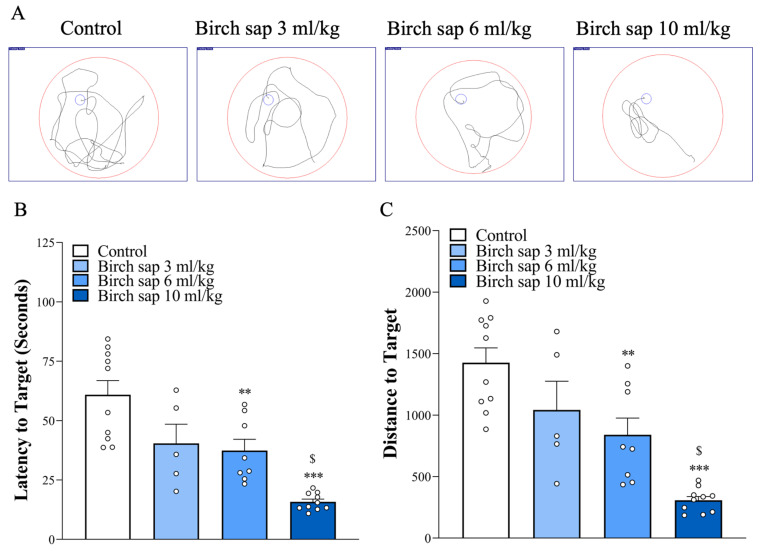
Spatial memory in the control and birch sap groups. (**A**) MWM trajectories, (**B**) time latency, and (**C**) distance traveled to reach the platform were assessed to evaluate spatial memory performance. One-way ANOVA with Tukey post-hoc test. ** *p* < 0.001, *** *p* < 0.0001 compared with the control group. ^$^
*p* < 0.0001 compared with the 6 mL/kg birch sap group. The red circle indicates the area of the swimming pool, and the white dot represents an ‘n’ value.

**Figure 4 ijms-26-05009-f004:**
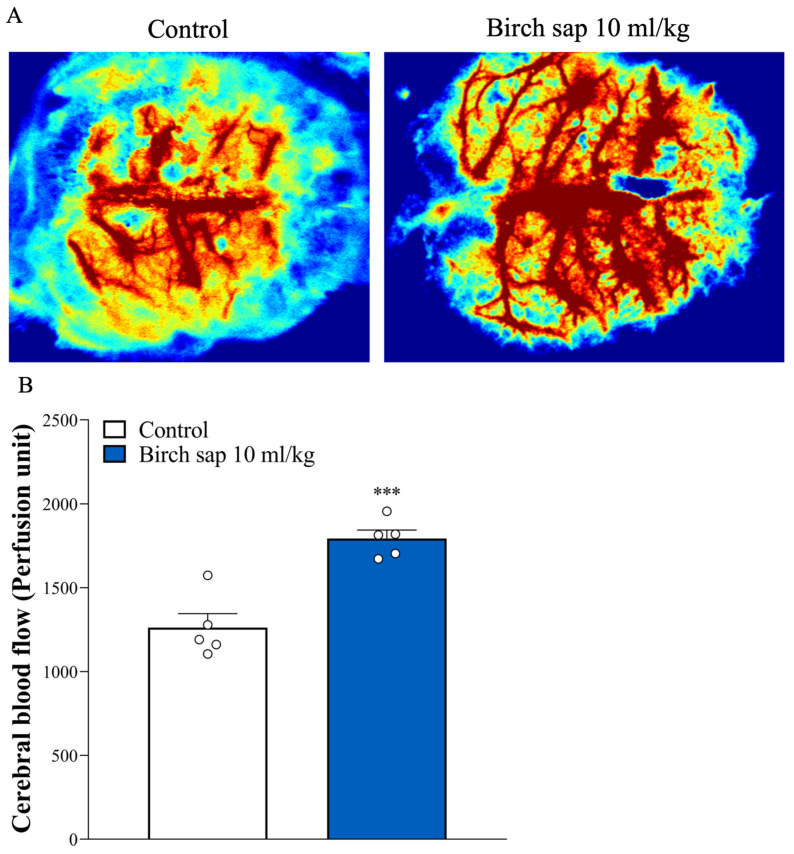
Cerebral blood flow in the control and birch sap groups. (**A**) Image of cerebral blood flow. (**B**) Perfusion units of cerebral blood flow. Student’s *t*-test. *** *p* < 0.0001 compared with the control group.

**Figure 5 ijms-26-05009-f005:**
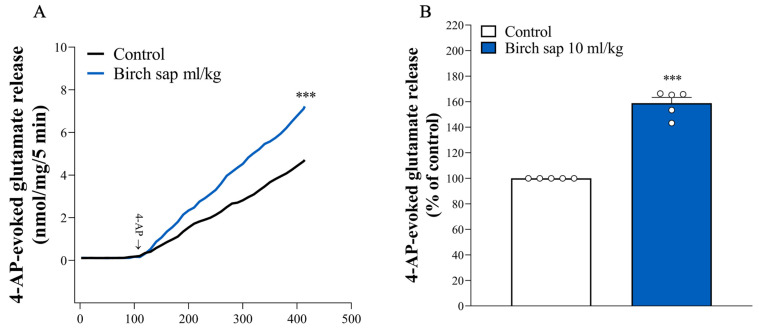
Glutamate release in the hippocampal synaptosomes of the control and birch sap groups. (**A**) The release of glutamate release was evoked by 1 mM 4-AP (arrow) in both the control and birch sap groups. (**B**) The results are expressed as a percentage of the 4-AP-evoked glutamate release (% of control). Student’s *t*-test. *** *p* < 0.0001 compared with the control group. The white dot represents an 'n' value.

**Figure 6 ijms-26-05009-f006:**
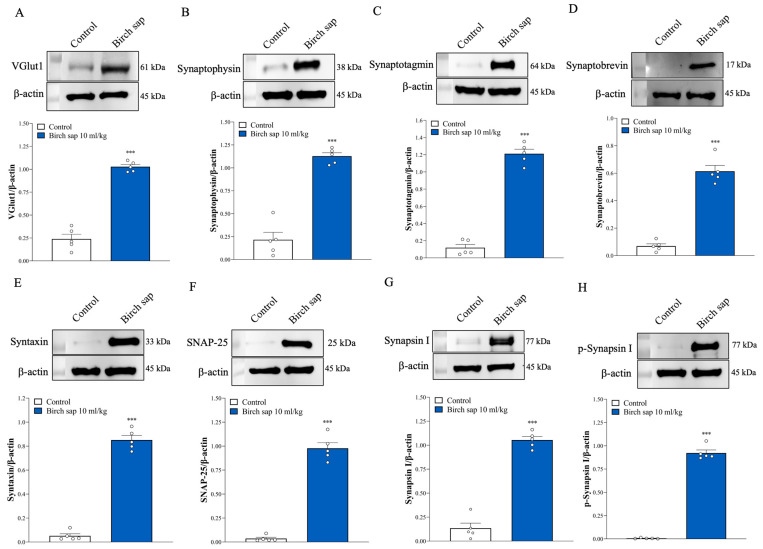
Protein levels of VGlut1, synaptophysin, synaptotagmin, synaptobrevin, syntaxin, SNAP-25, synapsin-1, and *p*-synapsin I in the hippocampal synaptosomes of the control and birch sap groups. (**A**–**H**) The image is cropped from the original data. Relative protein expression was normalized to β-actin. Student’s *t*-test. *** *p* < 0.0001 compared with the control group. The white dot represents an 'n' value.

**Figure 7 ijms-26-05009-f007:**
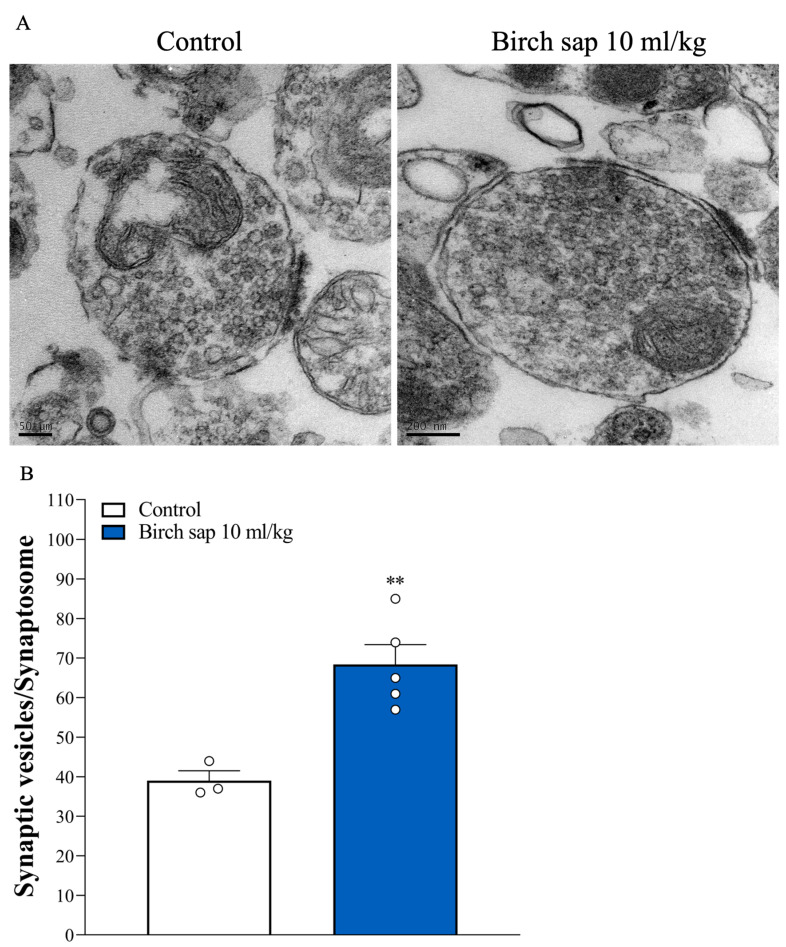
Synaptic vesicles in the control and birch sap groups. (**A**) TEM of hippocampal synaptosomes. Scale bar, 200 nm. (**B**) Quantification of the number of synaptic vesicles in hippocampal synaptosomnes. Student’s *t*-test. ** *p* < 0.001 compared with the control group. The white dot represents an ‘n’ value.

**Figure 8 ijms-26-05009-f008:**
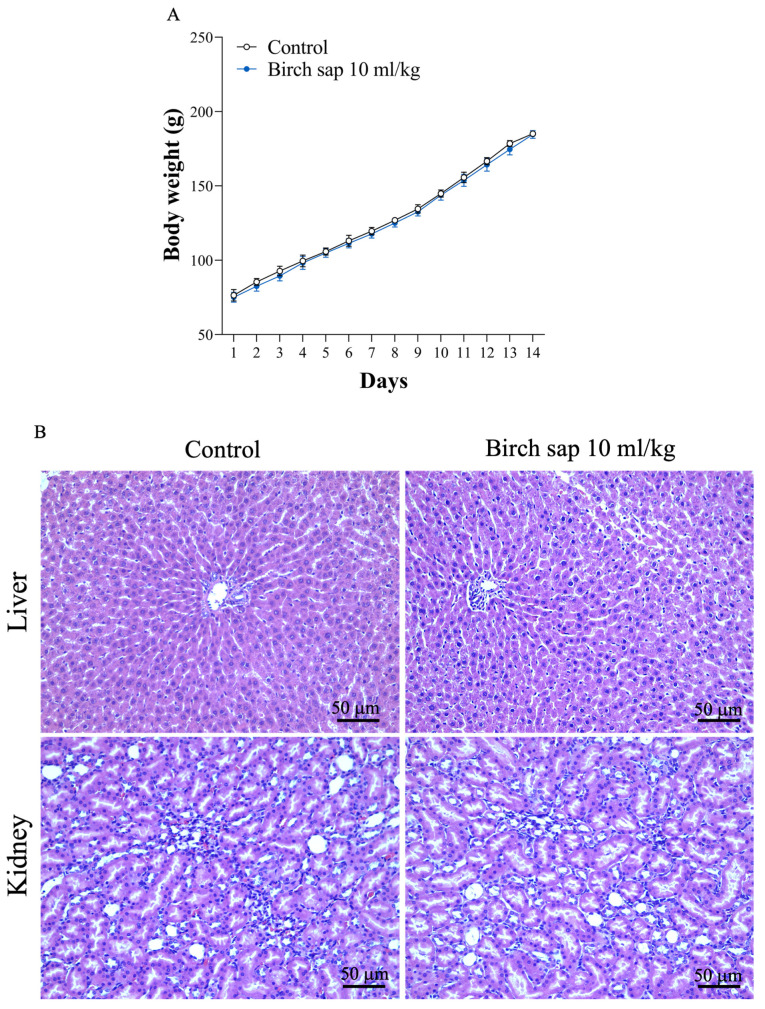
Rat body weight (**A**) and morphology of liver and kidney (**B**) in the control and birch sap groups. Liver and kidney morphology were examined using H&E staining at 400× magnification. Scale bars = 50 μm. Student’s *t*-test.

**Figure 9 ijms-26-05009-f009:**
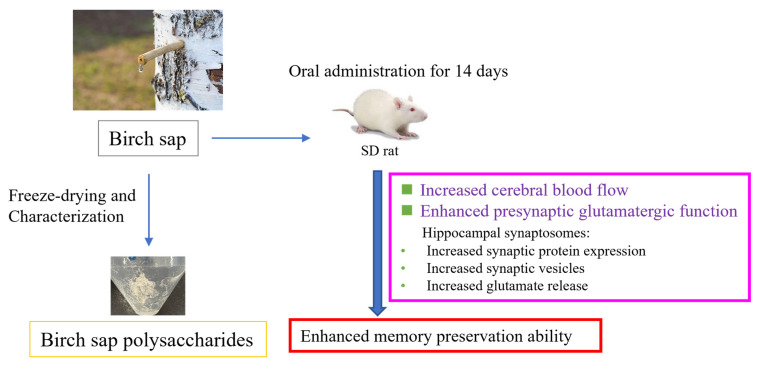
Graphical representation of the possible mechanisms of memory preservation induced by birch sap in rats. The effect of the oral administration of birch sap on memory may be due to the increased expression of synaptic proteins and a higher number of synaptic vesicles in the hippocampal nerve terminals, thereby enhancing glutamate release in the hippocampus of these rats. Additionally, memory preservation in the birch-sap-treated rats may be associated with its modulatory effects on cerebral blood flow.

**Table 1 ijms-26-05009-t001:** Monosaccharide composition analysis of birch sap.

	Monosaccharides (μmol/g Sample)
myo-inositol	1.16 ± 0.01
sorbitol	2.75 ± 0.01
fucose	628.95 ± 0.70
arabinose	4.66 ± 0.01
galactose	28.53 ± 0.01
glucose	1045.91 ± 0.67
mannose	53.70 ± 0.10
fructose	1693.67 ± 9.34

## Data Availability

The data presented in this study are available on request from the corresponding author.

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
