# Peer review of "Birch Sap Preserves Memory Function in Rats by Enhancing Cerebral Blood Flow and Modulating the Presynaptic Glutamatergic System in the Hippocampus"

_ijms, 2025, doi:10.3390/ijms26115009_

Round 1
Reviewer 1 Report
Comments and Suggestions for Authors
This study examined the effects of oral administration of birch sap on memory function in healthy rats to prevent or ameliorate memory impairment. 14 days after administration, the Morris Water Maze (MWM) test was performed, and changes in cerebral blood flow, synaptosomal neurotransmitter dynamics, and synaptic-related These studies are important for the prevention of memory impairment. While these studies may provide important guidance for the prevention of memory impairment, there are some points that need to be considered for improvement and discussion.
- First of all, we would like to point out that there are a number of typographical errors. The term "birch soap" is often used. This may be a typo, since it is different from the title, but the term "soap" ingestion seems incongruous, since "soap" is not originally ingested orally. Please rewrite it again to include this point.
- Since this is a short-term study limited to healthy rats and there are no studies in aged animal models or disease models (e.g., Alzheimer's disease model), we should be cautious about the application to humans.  
- I feel that the ingredients are not sufficiently specified. It is said to be "rich in polysaccharides," but it is unclear which components contribute to the effect, and identification of the active ingredients is necessary.
- There is a discrepancy between the method and the experimental results. The method states that only a control and a 10 mg dose group were created, but the results show results at various doses. These are not credible as experiments. It should be proven whether this is an error or falsification.
Author Response
First of all, we would like to point out that there are a number of typographical errors. The term "birch soap" is often used. This may be a typo, since it is different from the title, but the term "soap" ingestion seems incongruous, since "soap" is not originally ingested orally. Please rewrite it again to include this point.
The word "birch soap" in the manuscript is corrected to "birch sap".
Since this is a short-term study limited to healthy rats and there are no studies in aged animal models or disease models (e.g., Alzheimer's disease model), we should be cautious about the application to humans.
We thank the reviewer for the insightful comment regarding the limitations of our study. We acknowledge that the current investigation was conducted in healthy young adult rats over a relatively short duration. The absence of data from aged animals or disease models (e.g., Alzheimer’s disease models) indeed limits the translational applicability of our findings. For this limitation, the sentence in the discussion has been revised to " Thirdly, in the present study, birch sap was administered for a limited duration of two weeks. However, the long-term effects of birch sap ingestion remain unclear. Future stud-ies involving prolonged administration are warranted to evaluate potential behavioral or neurochemical alterations resulting from chronic exposure. Finally, other models of memory impairment, such as AD or aging animal models, should also be evaluated to enhance the clinical relevance of birch sap’s effects." (page 12, lines 250–255).
I feel that the ingredients are not sufficiently specified. It is said to be "rich in polysaccharides," but it is unclear which components contribute to the effect, and identification of the active ingredients is necessary.
We thank the reviewer for the valuable comments. We confirm that the active compound is indeed a low-molecular weight polysaccharide (MW 1.29 kDa). Due to its significant structural complexity and the limitations of our current research resources, a complete structural elucidation has proven challenging at this stage. However, we have provided key characterization data, including its monosaccharide composition (Fig. 1A & Table 1), molecular weight (Fig. 1B), and 1H NMR (Nuclear Magnetic Resonance) spectroscopic data (Fig. 1C) (page 2, lines 73-82). In addition, the sentence in the abstract has been revised to " Compositional analysis revealed that birch soap is rich in polysaccharides, specifically a low-molecular weight polysaccharide (MW 1.29 kDa)". Also, the sentence in the discussion has been also revised to " The observed preservation of memory function by birch sap may be attributed to its poly-saccharide content, as characterized by monosaccharide composition (Fig. 1A and Table 1), molecular weight (Fig. 1B), and 1H NMR spectroscopic analysis (Fig. 1C)." (page 11, lines 201-203).
While a definitive structure could not be fully resolved, these data are crucial for its preliminary identification and provide an important foundation for future, more in-depth structural studies.
There is a discrepancy between the method and the experimental results. The method states that only a control and a 10 mg dose group were created, but the results show results at various doses. These are not credible as experiments. It should be proven whether this is an error or falsification.
We thank the reviewer for pointing out this important issue. The discrepancy between the Methods and Results sections was due to an oversight. In the behavioral experiment, multiple dosage groups (3, 6, and 10 ml/kg) were used but not correctly described in the Methods. This has been corrected in the revised manuscript (page 13, lines 308–310).

Reviewer 2 Report
Comments and Suggestions for Authors
This paper is well written. The data is interesting and I ‘d like to know what components in birch sap are causing these effects.
The authors showed that birch sap ingestion increases synaptic granules in the brain and enhances glutamate release.
I have a few questions.
Young rats were used in this experiment, and they do not have decreased levels of glutamate. Since excess glutamate causes overexcitement, did the birch sap ingestion group experience overexcitement or fighting behavior between rats?
Alternatively, although the administration period in this experiment was 2 weeks, could the long-term ingestion of birch sap cause overexcitement?
Minor points
Line 181, References.
Author Response
This paper is well written. The data is interesting and I ‘d like to know what components in birch sap are causing these effects.
We confirm that the active fraction is rich in polysaccharides, or more specifically, a low-molecular weight polysaccharide (MW 1.29 kDa) (Page 2, lines 73-82). In addition, the sentence in the abstract has been revised to " Compositional analysis revealed that birch soap is rich in polysaccharides, specifically a low-molecular weight polysaccharide (MW 1.29 kDa)". Also, the sentence in the discussion has been also revised to " The observed preservation of memory function by birch sap may be attributed to its poly-saccharide content, as characterized by monosaccharide composition (Fig. 1A and Table 1), molecular weight (Fig. 1B), and 1H NMR spectroscopic analysis (Fig. 1C)." (page 11, lines 201-203).
The authors showed that birch sap ingestion increases synaptic granules in the brain and enhances glutamate release.
I have a few questions.
Young rats were used in this experiment, and they do not have decreased levels of glutamate. Since excess glutamate causes overexcitement, did the birch sap ingestion group experience overexcitement or fighting behavior between rats?
We thank the reviewer for this thoughtful comment. The sentence," Throughout the experimental period, all animals exhibited stable behavior and showed no overt signs of stress or agitation. ", has been added to the methods section (page 13; lines 310-311).
Alternatively, although the administration period in this experiment was 2 weeks, could the long-term ingestion of birch sap cause overexcitement?
We appreciate the reviewer’s insightful question. The sentence," Thirdly, in the present study, birch sap was administered for a limited duration of two weeks. However, the long-term effects of birch sap ingestion remain unclear. Future studies involving prolonged administration are warranted to evaluate potential behavioral or neurochemical alterations resulting from chronic exposure. ", has been added to the methods section (page 12; lines 250-253).
Minor points
Line 181, References.
This point is revised (page 11, line 181).

Round 2
Reviewer 1 Report
Comments and Suggestions for Authors
The authors have made appropriate corrections to my previous comments, and have also added MM and corrected typos. I will not be asking for any further corrections.